# The Performance of Shrinkage Estimator for Stock Portfolio Selection in Case of High Dimensionality

Nhat Minh Nguyen [1], Trung Duc Nguyen [1], Eleftherios I. Thalassinos [2,3] and Hoang Anh Le [4,*]

1   Faculty of Banking, Banking University HCMC, No. 36 Ton That Dam Street, Nguyen Thai Binh Ward, District 1, Ho Chi Minh City 700000, Vietnam; nhatnm@buh.edu.vn (N.M.N.); trungnd@buh.edu.vn (T.D.N.)
2   Faculty of Maritime and Industrial Studies, University of Piraeus, 185-33 Piraeus, Greece; thalassinos@ersj.eu
3   Faculty of Economics, Management and Accountancy, University of Malta, 2080 Msida, Malta
4   Institute for Research Science and Banking Technology, Banking University HCMC, No. 36 Ton That Dam Street, Nguyen Thai Binh Ward, District 1, Ho Chi Minh City 700000, Vietnam
*   Correspondence: anhlh_vnc@buh.edu.vn

**Abstract:** Harry Markowitz introduced the Modern Portfolio Theory (MPT) for the first time in 1952 which has been applied widely for optimal portfolio selection until now. However, the theory still has some limitations that come from the instability of covariance matrix input. This leads the selected portfolio from MPT model to change the status continuously and to suffer the high cost of transaction. The traditional estimator of the covariance matrix has not solved this limitation yet, especially when the dimensionality of the portfolio soars. Therefore, in this paper, we conduct a practical discussion on the feasible application of the shrinkage estimator of the covariance matrix, which is expected to encourage the investors focusing on the shrinkage–based framework for their portfolio selection. The empirical study on the Vietnam stock market in the period of 2011–2021 shows that the shrinkage approach has much better performance than other traditional methods on the primary portfolio evaluation criteria such as return, level of risk, Sharpe ratio, maximum loss, and Alpla coefficient, especially the superiority is even more evident when the dimension of covariance matrix increases. The shrinkage approach tends to create more stable and secure portfolios than other estimators, as demonstrated by the average volatility and maximum loss criteria with the lowest values. Meanwhile, the factor model approach is able to generate portfolios with higher average returns and lower portfolio turnover; and the traditional approach gives good results in the case of low—dimensionality. Besides, the shrinkage method also shows effectiveness when beating the tough market benchmarks such as VN-Index and 1/N portfolio strategy on almost performance metrics in all scenarios.

**Keywords:** shrinkage estimator of covariance matrix; shrinkage intensity; modern portfolio theory; portfolio management; high-dimensionality

## 1. Introduction

Covariance matrix estimation is one of the important approaches to improve the performance of portfolio selection, especially in the Markowitz model. As we all know, the sample covariance matrix does not really work well in the high—dimensionality, and in many cases when the matrix dimension is much larger than the sample size, the sample covariance matrix is singular (DeMiguel and Nogales 2009). This problem often occurs in the financial market when the number of investment securities in the market tends to be more and increase many times faster than the observed time.

To solve this problem, academic researchers and practitioners proposed many different ways for estimating the covariance matrix of the portfolio. For example, based on the single—index (SI) model introduced by Sharpe (1964), the practitioners can calculate the covariance matrix for their portfolio optimization. In the research of Senneret et al. (2016), there is ample evidence that the SI approach produces portfolios that are much less sensitive

to estimation error than the standard approach because it requires estimating only 2N + 1 parameters to construct the covariance matrix versus the N(N + 1)/2 parameters required by the standard approach and, therefore, perform better across a range of risk and return metrics. However, both the standard and SI approach to portfolio selection are vulnerable to estimation error from using the sample mean returns vector $m_T$ (SI is also exposed to estimation error from $\hat{\sigma}_m^2$). Elton et al. (2009) based on the constant correlation (CC) model that assumes that all stocks have the same correlation, equal to the sample (historical) mean correlation, to build the covariance matrix. This research found that the CC model is more accurate forecasts future covariance matrices than the sample covariance matrix and SI approach, from that producing portfolio with superior performance than competing models, across a range of metrics. However, the CC model still faces many problems in estimating a large dimensional covariance matrix for portfolio choice. Ledoit and Wolf (2003b, 2004) introduced a new approach to estimate the covariance matrix for portfolio selection, which is called the shrinkage method. The shrinkage of the covariance matrix is the weighted combination between the sample covariance matrix and the shrinkage target matrix with a stable structure (Corsaro et al. 2021; Sun et al. 2019). The approach has proven its effectiveness in portfolio selection in the case of high—dimensionality (Bodnar et al. 2021). DeMiguel et al. (2013) published a significant review paper which systematically described the shrinkage frameworks and related applications in practice.

To our knowledge, there is a gap for the investors to investigate the level of influence of covariance matrix estimators for portfolio selection, especially the robust estimator like shrinkage. The shrinkage approach may have proven its effectiveness in developed financial markets, but its applicability in the undeveloped or emerging financial market still needs further research. Therefore, in this paper, we conduct a practical discussion on feasible application of the shrinkage estimator of covariance matrix on the Vietnam stock market, from which it is expected to encourage the investors focusing on the shrinkage—based framework for their portfolio selection.

## 2. Literature Review

In general, the idea of a shrinkage estimator is to estimate $\hat{\Sigma}_{Shrink} = (1 - \delta_*)S_T + \delta_*\hat{\Sigma}_{target}$, where $\delta_*$ is determined by a data-driven algorithm and $\hat{\Sigma}_{target}$ is chosen through some prior belief about asset return covariance. In terms of shrinkage target matrices, Ledoit and Wolf (2003b) suggest a single—index model that relied on an important assumption that the return of the stock in the portfolio follows a factor—model structure, while Ledoit and Wolf (2003a) recommend a constant correlation model that based on the prior belief that all stocks have the positive average correlation. These researches showed that both of shrinkage estimators are superior to the traditional sample covariance matrix. However, the shrinkage to constant correlation model gives the best portfolio selection results when the size of the portfolio is less than 100 stocks, while the shrinkage to single index model has the best results if the portfolio size is larger than 225 shares. In addition, Ledoit and Wolf (2004) proposed a target matrix that does not have any characteristics of true covariance matrix like the two previous shrinkage methods. The target matrix is an identity matrix in which all the diagonal elements are one, and the remaining elements are equal to zero. With the introduction of this kind of shrinkage target matrix, Ledoit and Wolf tried to answer the question of whether an investor can choose an optimal portfolio if they are not using any the domain knowledge in the finance field.

In an important survey, Bai and Shi (2011) also summarized some of the nofigure contributions widely applied in the high-dimensional estimation of covariance matrix, such as shrinking, observable and implicit variables, even from Bayesian approach, and random matrix theory. Whereas, Yang et al. (2014) suggested a kind of hybrid approach of covariance matrix through the combination between robust M-estimation and a shrinkage calculation of Ledoit and Wolf (2004). On the other hand, Ikeda and Tatsuya (2016) found a class of generally weighted estimators involving a linear combination of sample covariance matrices with rule-based estimators and linear shrinkage estimators with no additional

and special factors under the component scheme. Konno (2009) proposed an estimation approach for large-dimensional covariance matrices having complex types of multivariate normal distributions when the size of the portfolio was greater than the number of observed samples. Unbiased risk estimates for certain groups of global covariance matrices were derived by considering these techniques under real and complex invariant quadratic forms of loss functions. In another scenario, Chen et al. (2010) adopted the shrinkage approach, and suggested an estimator of sample covariance relied on reducing the mean square error in Gaussian samples. In the recent time, the researchers also focused on non—linear shrinkage estimators that consider the eigenvalues distribution of covariance matrix instead of predicting the true covariance matrix (Ledoit and Wolf 2017a, 2017b, 2018). The non-linear shrinkage model has more flexible than the linear shrinkage of covariance matrix, but the long-running time and high computational cost are limitations of the non-linear approach (Lam 2016).

*Shrinkage Estimator of Covariance Matrix*

The overt advantage of shrinkage is able to take advantage of both a traditional sample covariance matrix and a shrinkage target matrix while avoiding their shortcomings. By figuring out an optimal shrinkage coefficient between two estimators, the estimated results can be more close to the true covariance matrix. In this paper, we use the single—index model as a shrinkage target matrix. Therefore, the shrinkage coefficient is calculated by the linear combination between the traditional sample covariance matrix and the covariance matrix of single—index model.

Ledoit and Wolf really succeeded in estimating the shrinkage coefficient based on Quadratic Loss function. The optimal shrinkage coefficient ($0 \leq \delta^* \leq 1$) is estimated as follows:

$$\delta_* = \frac{\sum_{i=1}^{N} \sum_{j=1}^{N} [Var(s_{ij}) - Cov(f_{ij}, s_{ij})]}{\sum_{i=1}^{N} \sum_{j=1}^{N} [Var(f_{ij} - s_{ij}) + (\theta_{ij} - \sigma_{ij})2]}$$

where in: $\sigma_{ij}$, $s_{ij}$ và $f_{ij}$ are components of true covariance matrix ($\sum$), sample covariance matrix (S), and shrinkage target matrix (F) with $\theta_{ij} = E(f_{ij})$ và $\sigma_{ij} = E(s_{ij})$.

## 3. Methodology

### 3.1. Input Data

To carry out all the experiments in this paper, we consider the weekly stock price, collected directly from the Vietnam stock market. After cleaning the data, we convert daily prices to weekly prices. Here, the weekly returns are computed from all stocks considered, and have been updated by dividends, and stock splits, for example. We consider the sample data S(t) = 520, referring to 520 weeks in the period of January 2011–January 2021. All the stock tickers must come from those firms listed on HoSE (Ho Chi Minh City Stock Exchange) for at least two years. This will eliminate companies that frequently enter or leave the market due to the failure to meet the criteria to get listed on the stock exchange, from that the considered portfolios will be more stable. Besides, the liquidity of these companies must be guaranteed, meaning that their daily trading volume must be greater than their average trading volume of the previous 20 days. Thus, the cumulative number of picked stock tickers, satisfy the requirement, is 350 companies in total. Also, it should be mentioned that the considered data is extracted from HoSE for quality assurance. Furthermore, the VN-Index, known as the performance of the Vietnam stock market, is taken into account as a reference index for shrinking scenario to a one—factor model.

### 3.2. Portfolio Performance Evaluation Methodology

In order to analyse the performance of the shrinkage estimator of the covariance matrix and other estimators, a back-testing system is built based on the research of Tran et al. (2020). The back-testing allows the (prior) statistical properties of the strategy to be examined, providing insight into whether a strategy can be profitable in the future.

Based on the back-testing system, a rolling-horizon procedure is applied to consider the different covariance matrix estimators for portfolio selection. In more details, beginning 1 January 2013, weekly historical data from two years back will be employed to calculate the covariance matrix parameter for the optimization procedure and to initialize the first portfolio. The period is called as in-the-sample. The period of January 2013–January 2021 shall be the evaluation dataset that is considered as out-of-sample. In addition, the rebalancing point of portfolios is on a weekly basis. The covariance matrix is recalculated, and the portfolio is optimized on the covariance matrix at every rebalancing point. This process will be repeated until the last rebalancing point. The reason why we choose the weekly frequency for testing because it is consistent with the characteristics of the Vietnamese stock market that the delay settlement date of a stock reach to some business days.

The metrics which are used for portfolio performance evaluation are similar the ones in the previous study of Nguyen et al. (2020), including portfolio's return, portfolio's risk, Sharpe ratio, maximum loss, winning ratio, portfolio turnover and Jensen's Alpha. In which, the criteria such as the portfolio's return, Sharpe ratio, winning ratio and Alpha coefficient show the ability to generate profit of the portfolios, whereas other criteria reflect the level of risk of the portfolios. In particular, the portfolio's return measures its gain/loss for a specific period of time; the Sharpe ratio considers the return of the portfolio compared with its risk; the winning ratio shows the number of the transaction having a positive return over total transactions; and the Jensen's Alpha identifies surpassed profit of portfolio compared with the expected rate of return calculated from Capital Asset Pricing Model (CAPM). While the portfolio's risk is defined as the volatility of portfolio return in a specific period; the portfolio turnover shows the portfolio stability at the time that a portfolio changes its status according to optimal strategy; and the maximum loss reflects a risk level of a portfolio under conditions of a bad and complicated market and is calculated from the highest value to the lowest value of a portfolio in a fixed period.

Moreover, transaction fees are also taken into account during back-testing when the stocks change their weights in the portfolio, so that the research results are consistent with real-world situations. The transaction fees will be 0.2% of the total transaction value, and this is a fee widely applied on the Vietnamese stock market. Last but not least, instead of simply checking the estimation methods on a single portfolio (N = 350 shares), the authors would check these estimation methods on four portfolios with specific stock numbers (N = 50, 100, 200, 350). The allocation of stocks into portfolios will be dependent on the market capitalization of those stocks. For example, N = 50 means that the portfolio will consist of 50 securities with the largest market capitalization; N = 100 is a portfolio of 100 securities with the highest market capitalization, equivalent to N = 200 and N = 350. The market capitalization of a company is measured by the number of shares outstanding multiplied by the trading price of the company.

### 3.3. Portfolio Selection

In this paper, we use the global minimum variance model (GMV) for selecting the optimal portfolio with the estimated covariance matrices. DeMiguel and Nogales (2009) suggested that as global minimum variance model (GMV) depends only on covariance matrices, so it is less vulnerable to estimation errors than conventional models. In addition, we also impose a limit on the weights of the stocks which are always larger than zero ($w_i > 0$). This means that we do not consider the case of short selling because this activity has not been widely developed in the Vietnamese stock market, especially during the research period. The optimal weights of stocks in portfolio is calculated as follows:

$$w_* = \Sigma^{-1}\mathbf{1}\left(\mathbf{1}^T\Sigma^{-1}\mathbf{1}\right)^{-1}$$

In which, we denote **1** for a vector of all ones, and $\Sigma$ is denoted for the estimated covariance matrix.

## 4. Result

### 4.1. Competing Estimators and Benchmarks

In addition to the shrinkage estimator above, we also consider the estimators of the covariance matrix as follows:

***Sample covariance matrix (SC)***

This is a square matrix that sets the sample covariance calculated from a data sample between any two stocks in the portfolio.

***Single—index model (SI)***

This is a one—factor covariance matrix of Sharpe (1964). It assumes that the returns of every asset in the market are impacted by the rest of the market. The covariance matrix of SI is estimated by the following formula:

$$\check{\sum} = \beta\beta^T\check{\sigma}_m^2 + \check{\sum}_\varepsilon$$

In which, $\check{\sigma}_m^2$ is the variance of the market portfolio; $\beta$ is the vector of coefficient determined from SI's regression with the length of N stocks.

To have a better understanding about the performance of the shrinkage estimator of the covariance matrix, we also need to evaluate its performance metrics with the benchmarks on the Vietnam stock market.

***Vietnam stock index (VN—Index)***

The VN-index, which is seen as one of the important indexes on the stock market, is a capitalization-weighted index of all the enterprises listed on HOSE. It is a relatively comprehensive measurement of the development from time to time of Vietnam's stock market, hence being used as a benchmark for the out-of—sample period.

***1/N portfolio***

It should be noticed that, the VN-Index uses weighted averages, indicating Vietnam's capital market is overwhelmingly dominated by large and state-owned companies, as well as controlled by certain industries. Therefore, we can use an equal weighted index by market capitalization—1/N portfolio as an alternative benchmark to reflect better what is happening in the economy. Moreover, DeMiguel and Nogales (2009) demonstrated that it is very difficult for an optimal strategy to outperform the performance of 1/N portfolio strategy. He applied 14 models across seven empirical datasets to select the optimized portfolios; however, the results showed that there is no better strategy than the 1/N strategy in term of portfolio performance metrics such as return, Sharpe ratio or turnover.

### 4.2. Empirical Results

The empirical results are presented in Table 1. These results show that portfolio selection based on shrinkage estimator of covariance matrix gives much better results compared with the traditional sample covariance matrix and the single—index covariance matrix in term of some criteria such as Sharpe ratio, portfolio's risk, maximum loss and Jensen's Alpha in almost sample groups (N = 100, 200, 350). However, regarding other evaluation criteria as portfolio's return, winning ratio, and portfolio turnover, the SI method is dominant over shrinkage method in all portfolio selection except for N = 50.

In particular, the mean of annual return by SI method in a testing period with four different portfolios gives superior results compared with shrinkage method and the average gap is about 1.65%, the greatest remarkable level is 3.15% when N = 200. Yet, in comparison with the SC method, the annual return gap between shrinkage and SC is in line with the increase of sample, it gains the highest gap which is 6.07% when N = 350. Thus, the shrinkage method can show its advantages in portfolio selection against with traditional SC but SI is still the optimal approach for this criterion.

**Table 1.** The performance of different estimators on out—of—sample from 1 January 2013–1 January 2021.

| Estimators | Portfolio Performance Metrics | | | | | | |
|---|---|---|---|---|---|---|---|
| | Portfolio's Return (Annually) | Sharpe Ratio | Jensen's Alpha | Winning Ratio | Portfolio's Risk (Annual Volatility) | Portfolio Turnover (Daily) | Maximum Loss |
| VN-Index | 13.00% | 0.6 | −0.18% | 55.31% | 17.03% | 0.10% | −45.34% |
| N = 50 | | | | | | | |
| 1/N portfolio | 9.55% | 0.45 | −1.41% | 56.01% | 15.64% | 1.03% | −41.63% |
| SC | 13.87% | 0.84 | 4.86% | 54.41% | 12.02% | 3.01% | −38.46% |
| SI | 13.46% | 0.73 | 2.52% | 56.66% | 14.04% | 1.12% | −36.35% |
| Shrinkage | 13.55% | 0.82 | 3.05% | 54.73% | 11.95% | 3.59% | −38.6% |
| N = 100 | | | | | | | |
| 1/N portfolio | 12.35% | 0.66 | 1.84% | 57.11% | 14.12% | 0.99% | −39.36% |
| SC | 11.93% | 0.8 | 4.38% | 54.26% | 9.86% | 4.11% | −27.64% |
| SI | 14.75% | 0.9 | 4.59% | 57.52% | 12.25% | 1.15% | −32.39% |
| Shrinkage | 12.69% | 0.89 | 5.06% | 54.81% | 9.63% | 3.65% | −25.81% |
| N = 200 | | | | | | | |
| 1/N portfolio | 15.53% | 0.98 | 5.79% | 59.02% | 12.09% | 1.02% | −31.47% |
| SC | 12.69% | 1.01 | 6.07% | 55.01% | 8.68% | 5.99% | −28.86% |
| SI | 17.43% | 1.26 | 8.08% | 59.52% | 10.48% | 1.11% | −25% |
| Shrinkage | 14.28% | 1.26 | 7.56% | 55.11% | 8.07% | 4.47% | −21.86% |
| N = 350 | | | | | | | |
| 1/N portfolio | 18.27% | 1.3 | 9.00% | 59.62% | 10.69% | 0.54% | −22.63% |
| SC | 13.95% | 0.85 | 4.89% | 55.01% | 11.9% | 2.81% | −38.76% |
| SI | 20.02% | 1.56 | 10.80% | 60.97% | 9.80% | 1.10% | −20.76% |
| Shrinkage | 18.57% | 1.85 | 11.84% | 57.87% | 7.48% | 4.38% | −16.73% |

The average level of risk or volatility of portfolio according to shrinkage approach also achieved the lowest value compared to the SI method and traditional SC method on four considered scenarios. More importantly, the bigger size of the portfolio is the lower volatility that the shrinkage method achieves. In Table 2, the volatility significantly decreases from 11.95% when N = 50 to 7.48% when N = 350 which show not only the stability of portfolio selection among different number sample of the portfolio but also reliability for risk-averse investors.

In connection with Sharpe ratio, due to the gap of volatility is higher than annual return gap when comparing between shrinkage method with SI and SC methods, the average Sharpe ratio of the portfolios by shrinkage method (1.2 times) is also better than the ones of SI (1.1 times) and SC (0.88 times) approaches. Similar to volatility, the higher number of stocks in the portfolio has, the higher Sharpe ratio that shrinkage release (1.85 times when N = 350), its high value also is preferable to the value by SI or SC as portfolio managers are able to foresee the trade-off between return and risk.

**Table 2.** The performance of the SC on out—of—sample from 1 January 2013–1 January 2021.

| Performance Metrics | The size of Portfolio (N) | | | |
|---|---|---|---|---|
| | **50** | **100** | **200** | **350** |
| Portfolio's return (Annually) |  |  |  |  |
| Sharpe ratio |  |  |  |  |
| Portfolio turnover (daily) |  |  |  |  |
| Maximum loss |  |  |  |  |

By considering the portfolio's daily turnover criteria, although the shrinkage method is less stable than SI method, the average daily turnover rate of shrinkage is 4.02% relevant to 20.1%/week much higher in comparison to the average daily turnover of SI which is just 1.12% relevant to 5.6%/week (Table 3). The change in daily turnover of the portfolio by SI is just from 1.1% to 1.15%, while portfolio by shrinkage method suffers significant change from 3.59% to 4.47% varied by the number of stocks in the portfolio. This accounts for the low annual rate of return by the shrinkage approach with its fairly high transaction fee according to the high daily turnover rate. In spite of unexpected results of daily turnover compared to the SI method, shrinkage method still has better outcomes against traditional SC method in terms of the daily turnover rate in almost portfolios.

The superiority of the profitability of SI method is presented by the overt results of the winning ratio on all considered portfolios. The winning ratio of the SI method is average value of 58.67% that is higher than 55.45% of shrinkage method and 54.67% of the SC method. By referring to the comparison between the shrinkage and SC method, the prevalence of the winning ratio by shrinkage method over the SC method increase in line with the number of stocks in the portfolio. The gap of the winning ratio between shrinkage and SC approach increase from 0.4% to 2.86% which proves that shrinkage is more excellent than the traditional SC in the criteria.

**Table 3.** The performance of the SI on out—of—sample from 1 January 2013–1 January 2021.

| Performance Metrics | The Size of Portfolio (N) | | | |
|---|---|---|---|---|
| | **50** | **100** | **200** | **350** |
| Portfolio's return (Annually) |  |  |  |  |
| Sharpe ratio |  |  |  |  |
| Portfolio turnover (daily) |  |  |  |  |
| Maximum loss |  |  |  |  |

In the context of unexpected financial happenings, the shrinkage method still brings more safety for investors than SI and SC method with absolutely outstanding results compared to SC and SI portfolios' maximum loss. In the period of 2013–2021, the shrinkage estimator does not show a clear difference from other estimators in the maximum loss (ML) criteria with the small size of portfolio (Table 4). However, when the number of stocks in portfolio soars, the ML criteria of shrinkage has improved significantly with a very low value of 16.73% compared to 38.76% of SC and 20.76% of SI. The average maximum drawdown on the four investment portfolios under consideration of the Shrinkage method is 25.7%, much lower than the average maximum loss of the SI method is 28.6% and the traditional SC method is 33.43%. Likewise, the maximum drawdown gap between shrinkage and both SI, SC method is the increasing trend in accordant with the number of stocks in the portfolio.

Notwithstanding inferior results of the annual return and winning ratio criteria, portfolios by shrinkage even have a fairly higher Jensen's Alpha value than the SI method in all scenarios. Especially in the high-dimensionality, Alpha coefficient of portfolios based on shrinkage approach has the highest positive value (11.84%) when N = 350. Moreover, the average value of Alpha coefficient of shrinkage method on four considered scenarios is about 6.9% pretty higher than that of SI and SC with the value of 6.49% and 5.05% respectively. This states that the shrinkage portfolio has earned the superior return than the theoretical portfolio return calculated from the CAPM because of either selection or timing skills, or both. However, this is not really true when the size of portfolio is small. In the experiment, the value of Alpha coefficient of shrinkage method is only about 3.05% that is much lower than the one of the traditional sample covariance matrix at 4.86% when N = 50.

**Table 4.** The performance of the shrinkage on out—of—sample from 1 January 2013–1 January 2021.

| Performance Metrics | The Size of Portfolio (N) | | | |
|---|---|---|---|---|
| | **50** | **100** | **200** | **350** |
| Portfolio's return (Annually) |  |  |  |  |
| Sharpe ratio |  |  |  |  |
| Portfolio turnover (daily) |  |  |  |  |
| Maximum loss |  |  |  |  |

In addition, when comparing to the benchmarks such as Vn-Index and 1/N portfolio, shrinkage-based portfolio also shows complete superiority over the benchmarks on almost the performance metrics in all scenarios. This superiority is more evident when the number of stocks in the portfolio tends to increase. Besides, the 1/N portfolio strategy is still a tough benchmark with the traditional estimator of covariance matrix. In the research, the performance of traditional SC method can not overcome the performance of the benchmark.

For shrinkage coefficient, its movement is mainly based on the size of the portfolio. When the number of stocks in the portfolio go up, the fluctuating area of shrinkage coefficient tends to increase. In Table 5, the fluctuating area of shrinkage coefficient are 0.05–0.3, 0.05–0.4, 0.1–0.5, and 0.1–0.6 corresponding to N = 50, 100, 200, 350. The higher value of shrinkage coefficient, the more it shows the importance of the shrinkage target matrix in determining the estimated covariance matrix for portfolio optimization because it is more involved in the estimation process of covariance matrix. In this case, the shrinkage target matrix has done its job well in minimizing the estimation errors arising in the traditional SC method, especially when N is high. In particular of N = 350, there are times when the shrinkage coefficient has reached very high values up to 0.6, which means that the shrinkage target matrix has affected 60% of the estimation of the covariance matrix. This explains why the performance of shrinkage estimator is better than that of other traditional estimators of covariane matrix in the high-dimensionality.

**Table 5.** The shrinkage coefficient in the period of 2013–2021.

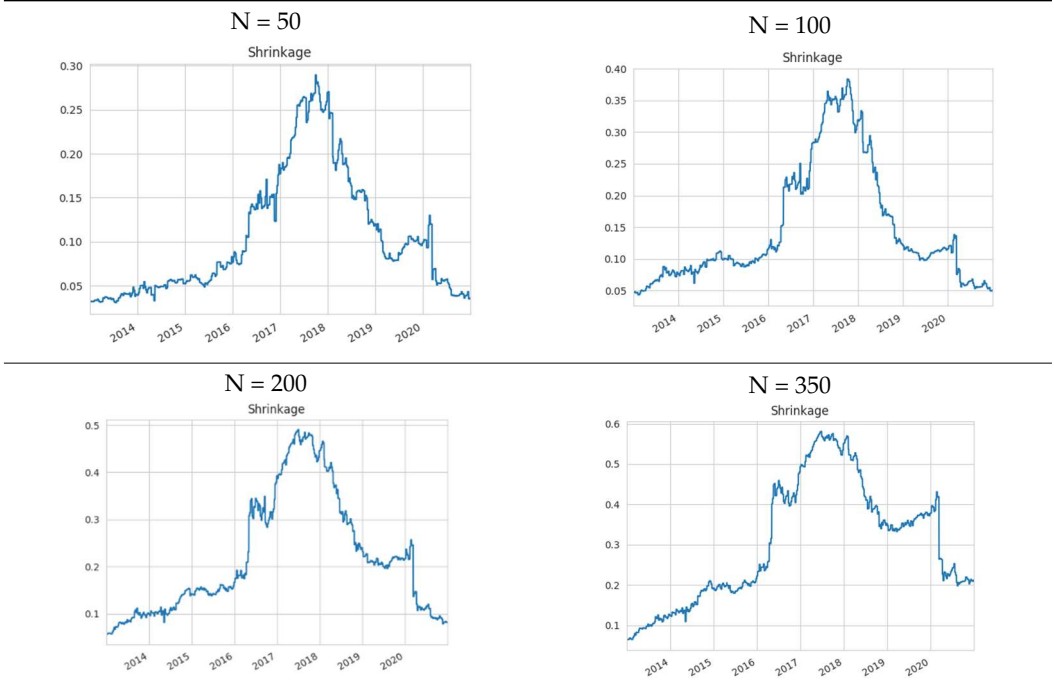

## 5. Conclusions

The empirical research results on the Vietnam stock market from 1 January 2011 to 1 January 2021 show that shrinkage estimator of covariance matrix has much better performance than the SI and traditional SC method. This superiority is reflected in most of the criteria used to measure the effectiveness of the portfolio including: the average volatility of the portfolio, Sharpe ratio, maximum loss and Jensen's Alpha. Besides, the shrinkage method also shows effectiveness when beating the tough market benchmarks such as VN-Index and 1/N portfolio strategy on almost the performance evaluation criteria in all scenarios.

The superiority of the shrinkage approach and the traditional SC method tends to increase as the number of shares in portfolio increases. This is explained by the inefficiencies in the processing capacity and many errors in estimating covariance matrix of traditional SC method when the dimension of the covariance matrix increases. Meanwhile, the shrinkage method tends to work better in the context of high number of shares in portfolio compared to SI method in term of significant criteria such as risk level, Sharpe ratio, maximum loss, or Alpha coefficient and even better than SC method with all above criteria plus annual return, winning ratio and portfolio turnover as well. This also implies very clear the ultimate purpose of portfolio selection with conservative strategy maximizing profit at the well-managed level of risk with huge number of stock in portfolio.

In addition, the shrinkage approach tends to create more stable and secure portfolios than other estimators, as demonstrated by the average volatility and maximum loss criteria with the lowest values. This is even more evident in the high-dimensionality. Meanwhile, the SI method is able to generate portfolios with higher average returns and lower portfolio turnover; and the traditional SC approach gives good results in the case of low—dimensionality.

In this research, the authors only consider the dimension of covariance matrix with the maximum number of shares N = 350, in the future researchers may increase the considered number of shares to better assess the difference between the estimators of covariance matrix in the selection of optimal portfolio on Vietnam's stock market. In addition, the researchers may consider changing other shrinkage target matrix instead of using one-factor model. Furthermore, they can change the way of linear combination in the shrinking method by

other non-linear techniques to create new covariance matrices with better stability and prediction in the optimal portfolio selection.

**Author Contributions:** T.D.N. conceived the idea, wrote the Introduction. H.A.L. wrote Literature review, Methodology. N.M.N. wrote Empirical results, Conclusion. E.I.T. revised the manuscript. All authors have read and agreed to the published version of the manuscript.

**Funding:** The study was supported by The Youth Incubator for Science and Technology Programe, managed by Youth Development Science and Technology Center—Ho Chi Minh Communist Youth Union and Department of Science and Technology of Ho Chi Minh City, the contract number is "12/2020/HĐ-KHCNT-VU".

**Institutional Review Board Statement:** Not applicable.

**Informed Consent Statement:** Not applicable.

**Data Availability Statement:** The data used to support the findings of this study have been deposited in the Github repository (https://github.com/anhle32/Shrinkage-estimator-of-covariance-matrix.git) (accessed on 23 September 2021).

**Conflicts of Interest:** The authors declare no conflict of interest.

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
