# Peer review of "The Performance of Shrinkage Estimator for Stock Portfolio Selection in Case of High Dimensionality"

_jrfm, doi:10.3390/jrfm15060249_

Round 1
Reviewer 1 Report
It is well known that the sample covariance matrix is affected
by estimation error; this is particularly severe in large portfolios.
THe application of shrinkage estimator permits to stabilyze the problems and it is widly used in Markovitz framework.
The authors analyze the use of shrinkage on Vietnam stock market in the period of 2011 – 2021.
I think that, although there is no theoretical or methodological contribution, the their empirical study Vietnam stock market
could improve the state of the art.
My suggestion is to accept the paper for publication, provided the authors are
willing to address the following suggestions.
Minor comment
- I think that the authors must specify that the use of shrinkage estimator is widly used in Markovitz framework, citing also more recent references
See for example:
1. Sun, Ruili and Ma, Tiefeng and Liu, Shuangzhe and Sathye, Milind, "Improved Covariance Matrix Estimation for Portfolio Risk Measurement: A Review",bvJournal of Risk and Financial Management, 2019, 12(1), 48;
2. S. Corsaro, V. De Simone, Z. Marino, "Split Bregman iteration for multi-period mean variance portfolio optimization", Applied Mathematics and Computation, 2021, 392, 125715;
3. T.Bodnar, Y. Okhrin, and N.Paroly, "Optimal Shrinkage-Based Portfolio Selection in High Dimensions", JOURNAL OF BUSINESS & ECONOMIC STATISTICS, 2021, AHEAD-OF-PRINT, 1-17
- There are typos that should be corrected (an automatic spelling checker is recommended) and grammar mistakes
Author Response
Many thanks for the support of the reviewer. Please see the attachment.

Reviewer 2 Report
see letter

Author Response

(The authors gave the same response as above.)
